# Biological and Oxidative Degradation of Ultrathin-Fibrous Nonwovens Based on Poly(lactic Acid)/Poly(3-Hydroxybutyrate) Blends

**DOI:** 10.3390/ijms24097979

**Published:** 2023-04-28

**Authors:** Anatoly Aleksandrovich Olkhov, Elena Evgenyevna Mastalygina, Vasily Andreevich Ovchinnikov, Alexander Sergeevich Kurnosov, Anatoly Anatolyevich Popov, Alexey Leonidovich Iordanskii

**Affiliations:** 1Scientific Laboratory “Advanced Composite Materials and Technologies”, Plekhanov Russian University of Economics, 36 Stremyanny Ln, 117997 Moscow, Russia; elena.mastalygina@gmail.com (E.E.M.);; 2N.N. Semenov Federal Research Center for Chemical Physics, Russian Academy of Sciences, 4 Kosygin St., 119334 Moscow, Russia; 3Institute of Biochemical Physics named after N.M. Emanuel, Russian Academy of Sciences, 4 Kosygin St., 119991 Moscow, Russia

**Keywords:** poly(lactic acid), poly(3-hydroxybutyrate), polymer blend, non-woven matrix, electrospinning, ultrathin-fibrous nonwovens, biodegradation, oxidation by ozone

## Abstract

Developing biodegradable materials based on polymer blends with a programmable self-destruction period in the environmental conditions of living systems is a promising direction in polymer chemistry. In this work, novel non-woven fibrous materials obtained by electrospinning based on the blends of poly(lactic acid) (PLA) and poly(3-hydroxybutyrate) (PHB) were developed. The kinetics of biodegradation was studied in the aquatic environment of the inoculum of soil microorganisms. Oxidative degradation was studied under the ozone gaseous medium. The changes in chemical composition and structure of the materials were studied by optical microscopy, DSC, TGA, and FTIR-spectroscopy. The disappearance of the structural bands of PHB in the IR-spectra of the blends and a significant decrease in the enthalpy of melting after 90 days of exposure in the inoculum indicated the biodegradation of PHB while PLA remained stable. It was shown that the rate of ozonation was higher for PLA and the blends with a high content of PLA. The lower density of the amorphous regions of the blends determined an increased rate of their oxidation by ozone compared to homopolymers. The optimal composition in terms of degradation kinetics is a fibrous material based on the blend of 30PLA/70PHB that can be used as an effective ecosorbent, for biopackaging, and as a highly porous covering material for agricultural purposes.

## 1. Introduction

Natural and synthetic biodegradable polyesters such as poly(lactic acid) (PLA), poly(3-hydroxybutyrate) (PHB), and copolymers based on them are a modern alternative to traditional non-degradable polymers in the manufacturing of disposable materials and products for medicine, packaging, agriculture, and environmental protection [1]. These biopolymers have satisfactory mechanical characteristics close to those of synthetic polymers and could be processed by traditional equipment [2,3]. After the end of their service life, materials based on PLA and PHB could be utilized under industrial compost conditions with the formation of environmentally friendly substances [4]. In line with other high-molecular compounds based on α-hydroxy acids and their derivatives, PLA and PHB have an ability to cause hydrolytic degradation due to the presence of ester groups [5].

Using blends of PHB and PLA to obtain packaging materials and sorbents with a bactericidal effect and an ability to accelerate biodegradation is a promising direction in developing novel polymer composites with a valuable set of properties [6]. The good water vapor and oxygen permeability of PLA and PHB make them appropriate for usage in packaging materials for fresh products [7]. The moderate hydrophobicity and high specific surface area of non-woven fibrous materials based on them allow for the creation of highly efficient sorbents for oil and oil products in the elimination of environmental disasters [8]. In this case, the materials are exposed to an aggressive action of moisture, oxygen, ozone, and other environmental factors, leading to the destruction of the polymer structure.

An important aspect of the development of polymer blends with specified properties is the possibility of regulating intermolecular interactions. To improve the compatibility of PHB and PLA and thus the complex of physical–mechanical and rheological parameters, two different types of compatibilizers were used in the work [9]: an ethylene oxide/propylene oxide block copolymer and a mixture of two liquid surfactants with a variable lipophilic–hydrophilic composition. Liquid surfactants were more effective than solid copolymers in improving morphology, as evidenced by the results of rheological measurements. In addition, thermal analyses showed that the presence of both types of compatibilizers causes an increase in the degree of crystallinity of the mixtures, which led to a noticeable increase in the values of the elastic modulus for the compatibilized mixtures compared to the initial ones.

In a study [10], to increase the relative elongation (by up to 16%), up to 6% of graphene oxide (GO) and cinnamaldehyde as a nanoreinforcing component were introduced into the blends of PHB and PLA using supercritical CO_2_. The disintegration ability of the developed materials decreased with the introduction of GO from 14 to 23 days without deterioration in the biodegradability characteristics of the final material.

Arrieta M.P. et al. [11] found that the addition of PHB to the PLA matrix significantly reduced the formation of bulbs on the PLA/PHB flexible fibrous materials. PHB in the blends acted as a nucleating agent for PLA. FTIR and Raman studies showed an interaction between PLA and PHB, mostly in a 75:25 ratio. PLA and 75PLA/25PHB fibers were additionally plasticized with acetyl tri-n-butyl citrate (ATBC), which resulted in an increase in elongation at break. The 75PLA/25PHB/ATBC ternary system showed improved thermal stability due to the good interaction between the components. Degradation tests under composting conditions confirmed the good biodegradability of all fibrous material formulations. It was found that PHB in some measure slows down the decay, while ATBC accelerates it. In the next work, Arrieta M.P. et al. [12] introduced lactic acid oligomer in PLA/PHB blends and obtained fibrous materials by electrospinning. It turned out that small (up to 20%) amounts of the oligomer improve the interaction between polymers, increasing the rate of crystallization. At the same time, the viscosity of the solutions as well as the fiber diameter decreased. Over 20% of oligomer caused the appearance of thickening on the fibers and the decrease in physical and mechanical properties.

Li X. et al. [13] demonstrated an integrated process solution: a low-shear, low-flow-rate, and melt-quenching process, which was used to produce high-performance PLA/PHB composites with optimal properties. Low shear minimizes molecular weight loss during heat treatment, providing optimum mechanical properties. In addition, the process of melt quenching made it possible not only to preserve the original fibrillar structure of PHB, but also to significantly increase the crystallization rate of the PLA phase, which led to the production of highly crystalline PLA/PHB mixtures. This technology significantly increased the rate of decomposition of PLA in mixtures, providing a weight loss of 90.9% in 10 days of decomposition in an alkaline solution.

Modification of PLA/PHB blends with microcellulose fibers is discussed in the paper [14]. It was shown that the introduction of cellulose microfibers increased the thermal stability and the dynamic mechanical parameters of the blends.

The studies of composites based on PLA and PHB were carried out in a number of works [15,16,17], where the influence of ultraviolet radiation, oxygen, and soil microbiota on the materials’ structure was studied, but the detailed mechanism of degradation on morphology remains insufficiently understood at present.

Since PLA biodegrades quite slowly under natural conditions, blends with PHB were obtained and studied. PHB has a number of advantages, one of which is its high biodegradation. The choice of PLA was justified by its relatively low cost and a wider range of applications in various fields of human activity. In addition, PLA improves the technological and mechanical properties of materials. Given the rather low hydrophilicity of PHB and PLA, these materials adsorbed oil to a better extent than water. This was the reason for the high selectivity of PLA/PHB nonwoven fibrous materials.

The study of the degradation of fibrous materials based on the blends of PHB and PLA is a continuation of our previously published works [18]. These materials demonstrated high sorption capacity in relation to oil products. Therefore, their intended use is in the field of environmental protection as environmentally friendly biopolymer sorbents capable of rapid biodegradation in natural conditions.

In this work, fibrous materials based on PLA/PHB blends of different composition were justifiably chosen as objects of study, as they have clear advantages over films: high specific surface area, high porosity, low thermal conductivity, high permeability, etc. These characteristics allow non-woven fibrous materials to compete successfully with films when used as sorbents, filters, agricultural materials, geotextiles, packaging, medical supplies, and for tissue engineering [19]. The materials under investigation were obtained by the electrospinning technique, which allowed for obtaining strictly oriented materials based on biopolymers with various highly functional additives (such as silk fibroin) and for obtaining matrices with high mechanical properties and a proliferation rate of living cells [20]. Nitric oxide donor substances can even be used as a functional substance to improve wound healing [21]. The unique properties of ultrathin fibers and nanofibers obtained from biopolymers, their mixtures, and functional modifications open up a significant potential for these biocompatible, biodegradable, environmentally friendly materials in the field of biomedicine, healthcare, packaging, and environmental protection. The structural modification of biopolymers (blends, composites, nanoparticles, copolymers, functional low-molecular-weight additives, plasticization) allows for overcoming various problems in technology and the application of non-woven fibrous biopolymer materials [22].

The aim of this work was to develop ultrathin fibrous matrices based on PLA/PHB blends by the electrospinning method and investigate the processes of biological degradation under soil microorganisms and oxidation by ozone using spectroscopical and structural–dynamic methods.

## 2. Results and Discussion

Non-woven fibrous materials can be successfully used as disposable, environmentally friendly, biodegradable products; sorbents for cleaning water or soil from oil products [23] or heavy metal compounds [24,25]. An important property of these disposable materials is their ability to have programmable degradation under environmental conditions: microorganisms, water, and oxidizing agents. The Sturm test (soil medium inoculum) provides the most realistic indication of the biodegradability of materials by the total soil microbiota. Evaluation of the ability of materials to experience oxidative degradation under the action of ozone, in turn, allows us to evaluate the possibility of recycling materials under the influence of abiotic environmental factors. Ozone was chosen as an oxidizing agent as a rather aggressive natural factor.

Figure 1 shows the kinetic curves of the degree of biological decomposition (according to the amount of carbon dioxide released during the mineralization of organic matter). It can be seen that the fibrous matrices based on PHB and 10PLA/90PHB are largely subject to biodegradation. At the same time, it is noticeable that the addition of even 10 wt.% of PLA significantly reduces the rate of biodegradation. When the content of PHB is below 10 wt.%, the degree of biodegradation of the samples does not exceed 3%. It is clearly seen that samples enriched with PHB undergo significant biodegradation. It can be noted that the defragmentation of the blends begins at a PHB content of more than 70 wt.%. It can be assumed that, in this concentration range, a continuous PHB phase is formed in the fiber, while the PLA phase becomes discrete.

It can be assumed that the biodegradation (enzymatic hydrolysis) is predominantly applied to the PHB phase, while PLA is practically not subjected to the degradation. Apparently, PLA hydrolysis products (lactic acid) are poorly assimilated by soil microflora and are an inhibitor of probiota. PHB, on the contrary, is actively hydrolyzed and consumed by soil microflora.

Figure 2 demonstrates the images of the samples after 90 days’ incubation compared to the initial materials.

This is clearly seen in the micrographs of PLA/PHB (10/90 wt.%) and PHB in the dependence of the degree of biodegradation on time (Figure 3). According to the results of microscopic analysis, the initial non-woven materials based on PLA and mixtures with PLA content from 30 to 90 wt.% were microfiber materials with an average fiber diameter of 8–10 μm. The average fiber diameter was practically independent of the ratio of polymers in the blends. This parameter is strongly influenced by the molecular weight of the polymer, the viscosity and electrical conductivity of the solution, and the type of solvent. For example, in [26], the average diameter of PLA/PHB mixed fibers when using a mixture of DMF/chloroform solvents was 1–2 µm. The fibers have a regular cylindrical shape. Almost no structural defects were observed. At the same time, the structure of the material based on pure PHB and a mixture of 10PLA/90PHB was characterized by defects; the fibers had numerous elliptical thickenings. The discovered fact indicates the effect of PLA on the technological properties of PHB during electrospinning molding.

After testing by the Sturm method, samples based on mixtures with a high content of PLA (70–100 wt.%) retained their integrity. This indicates the encapsulation of PHB by the PLA phase during the formation of microfibers. In mixtures with a higher content of PHB (70–90 wt.%), multiple focal areas of the hydrolyzed PHB phase were clearly visible, and are structures of the “adhesion” type with included sections of the PLA phase in the form of individual fiber fragments. PHB fibrous material in the process of biodegradation completely loses its original geometry and turns into a shapeless mass.

The DSC method was used to analyze the state of the supramolecular structure of the fibers. Figure 4 shows heating thermograms, and Table 1 shows temperatures and enthalpies of phase transitions in nonwoven fibrous materials before and after biodegradation. When analyzing thermograms, the stability of the PLA phase should be determined. It should be noted that, in mixtures with 30 wt.% PHB, the y of the PLA phase noticeably decreases, and, with a further increase in the PHB content, the exothermic peak of “cold” crystallization completely disappears. This can indicate a weak intermolecular interaction of polymers. PHB fibers and mixtures with a high PHB content undergo hydrolysis, as a result of which PHB is almost completely degraded. On thermograms with a high content of PHB, the crystalline phase of the latter disappears completely or significantly decreases. This is indicated by a decrease in the enthalpies of melting in proportion to the content of PHB in the blends.

An analysis of tabular data on the temperatures and heats of melting of polymer phases fully confirms the above. After biodegradation, the melting temperature of the crystalline phase in mixtures decreases and the crystallinity (enthalpy) drops more significantly both during the first heating and during the secondary one.

It should be noted that the heat of fusion of the crystalline phase in the initial mixtures decreases in direct proportion to the content of the components. This may indirectly indicate a slight interaction between PLA and PHB in blended fibers. The drop in the heat of fusion of PHB after degradation during secondary heating may indicate oxidative degradation of the polymer. However, in mixtures with PLA, this effect is not observed. More inert to microbiota and oxidation, PLA works as a PHB stabilizer [27].

DSC data fully confirm the study of fibrous materials by IR spectroscopy. The spectra of materials before and after degradation are shown in Figure 5. It can be seen that the bands characteristic of PLA remain practically unchanged after biodegradation, while the bands of PHB almost completely disappear.

If we considered the chemical structure of the polymers, we would expect a more significant intermolecular interaction between the methyl and weakly polar ester groups of PLA, as well as the end hydroxyl groups of PHB. To analyze the presence and level of intermolecular interaction between PLA and PHB, TGA thermograms were obtained. Figure 6 shows the TGA data.

It was shown that the PLA phase is more thermostable. The temperature of complete thermal degradation of HGB is around 300 °C, and for the PLA phase is around 375 °C. For the blends, we observe the additive dependence of thermal degradation of less thermally stable PHB on its concentration in the fibers. That is, polymers practically do not interact at the interface. Otherwise, the thermal behavior of the blends would vary with the composition and would depend on the contact area between the phases and the extent of the interfacial layer.

This peculiarity of the structure of blended PLA/PHB fibers should also appear during oxidation. As is known, ozone is an extremely aggressive oxidizing agent that has a marked effect on the chemical structure and molecular dynamics of polymers. However, the effect of ozone and soil inoculum on the morphology and dynamic characteristics of blends of biodegradable PLA and PHB polyesters remains insufficiently studied at present.

As is known, the structure of amorphous regions of crystallizing polymers is largely determined by the amount and the state of the crystalline phase. During formation of PLA/PHB mixtures, the structure and molecular dynamics of amorphous regions at the polymer phase interface change due to a decrease in the degree of crystallinity of PHB and PLA.

For a more complete characterization of the effect of ozone on the structure of the fibers, the fraction of absorbed gas was recorded. For each sample, the amount of absorbed ozone was determined by a calculation method. Based on the values of the absorbed gas volume, a number of dependencies between the material structure and the ability of samples to react with free ozone, which was supplied to the gas chamber where the sample was installed (Figure 7), were established.

Ozonation of polyesters occurs in several directions. At least two sequential processes that influence their morphology and molecular dynamics can be distinguished. First, there is the formation of a three-dimensional mesh due to the interaction of ozone with the side groups of polyesters located in the intracrystalline polymer space, where the elongated polymer chains are in close contact with each other. As a result of the conformational restriction for the straightening segments, the segmental mobility decreases, which leads to a decrease in the ozone diffusion coefficient. The second stage of the process, occurring upon further exposure to ozone, is the degradation of the polyether chain. In particular, it was shown in [13] that PHBs have a decrease in average molecular weight due to the break of the main chain, which leads to an increase in segmental mobility due to the removal of conformational restrictions.

To estimate the ozone consumption of PLA and PHB homopolymers and their mixtures formed in the process of electroforming in the form of ultrafine fibers, kinetic curves were obtained by a previously developed method in a flow reactor under steady-state conditions. Figure 8 shows the time dependence of ozone absorption by fiber mixtures of PLA/PHB with different mass ratios of components.

The two-step nature of ozone uptake most likely reflects sequential reactions of both types: ozonation of side groups leading to the formation of side products (A) and destruction of macromolecules accompanied by chain breaking (B). The first stage of ozonolysis proceeds according to the quasi-zero-order reaction, which agrees with the results of [15]. The second stage of the reaction, which consists in the destruction of the main chain, has a different-from-zero-order reaction.

As can be seen from the data shown in Figure 8, the absorption curves of ozone by polylactide and mixed PLA/PHB fibers of different composition, except for absorption by poly-(3-hydroxybutyrate), have the same appearance, namely, about the first 40 min the kinetic curves are satisfactorily described by a linear equation, and then their shape changes appreciably due to the increasing depth of ozonolysis. The exception is the PHB, in which there is no inflection point on the curve for the registered ozone flow rate and the overall kinetic profile of O_3_ absorption is monotonically increasing.

The dependence of the rate constant of ozonolysis on the PHB content has a clear maximum at the equivalent ratio of PHB and PLA. The higher rate of O_3_ absorption by the mixed fibers of PLA/PHB as compared to the absorption rate of homopolymers is explained by the decrease in the overall crystallinity of the blends, especially the crystallinity of PHB occurring in the process of mixing. As a consequence, the accessibility of the amorphous regions of the polymers to ozone increases. In fact, as was shown earlier, for fibrillar and film PLA/PHB composites [16], crystallinity decreases in a certain range of polyester ratios, which is associated with an increase in segmental mobility.

The intensity of probe rotation only in the amorphous space depends on the ratio of PLA/PHB in the fibers.

In the process of electroforming, under the action of electrodynamic and viscoelastic forces, the jet of mixed polymers in the gel state represents macromolecules, which are partially oriented along the fiber drawing axis. During the curing of the jet and the subsequent transition to the glassy state, the PHB and PLA molecules incompletely lose their orientation, and most of the macromolecules retain their still-forced elongated state. The phenomenon of macromolecule orientation when exposed to ozone was previously discovered in [28]. In this case, in two-phase polymeric systems, there is a forced separation of polymeric components with the formation of an additional free volume. The properties of such compositions intended for oil sorption were studied in [23,24].

At deeper degrees of ozonation of blended fiber matrices, some increase in correlation times is observed, indicating compaction of amorphous regions occurring, probably, as a result of formation of intermolecular cross-links in polyesters. During the oxidation of polyesters, their stiffness may increase due to the formation of additional polar groups and subsequent physical and chemical intermolecular cross-linking (e.g., with the formation of hydrogen bonds) accompanied by the loss of segmental mobility.

Ideal segmental packing is more characteristic of homopolymers and is practically not realized in binary mixtures due to poor compatibility of PHB and PLA, which is confirmed by DSC data. The lower packing density in the amorphous regions of the blended samples as compared to the homopolymers determines their increased rate of ozone oxidation. It should be noted that the ozonation rate in PLA/PHB mixtures depends on the segmental mobility of the polymer main chain in amorphous regions, which decreases as a result of a small increase in the intermolecular interaction at the polymer phase interface in the inversion region.

## 3. Material and Methods

### 3.1. Materials

The objects of investigation were non-woven matrices based on the blends of poly(3-hydroxybutyrate) (PHB) Lot 16F, supplied by BIOMER (Germany), and poly(lactic acid) (PLA) Ingeo 3801X Injection Molding Grade, supplied by NatureWorks (Minneapolis, MN, USA). The characteristics of PHB are as follows: a powder form obtained by microbiological synthesis with a molecular weight of 460 kDa, a melting point of 175 °C, and a density of 1.25 g/cm^3^. The characteristics of PLA are as follows: a granulated form with a viscosity average molecular weight of 1.9 × 10^5^ g/mol, a melting point of 166–173 °C, a glass transition temperature of 45 °C, and a density of 1.33 g/cm^3^. The PLA content in the blend was 0, 10, 30, 70, 90, and 100 wt.%.

The polymers’ concertation in the molding solutions and electrospinning parameters were fixed according to the previous results. In the work [29], the issue of using electrolytes in the PHB molding solutions was analyzed in sufficient detail. In subsequent works, we abandoned the use of electrolytes since they negatively affect the molecular parameters of the polymer [30]. It should be noted that the technology of electrospinning of the solutions of PLA/PHB blends did not differ much from the electrospinning of PHB solutions.

The solutions of the applied polymers were prepared using chloroform. Chloroform was chosen as a common solvent for PHB and PLA because of its relatively low toxicity and high desorption rate. The total concentration of polymers in the solutions was 7 wt.%. The concentration of polymers in the solution was chosen based on the stability of the electrospinning process. The viscosity range of mixed solutions was 2–5 Pa×s. For PHB solutions, depending on the concentration of 5–9 wt.%, this parameter varied in the range of 1.5–4.0 Pa×s. The optimum performance and stability of the process was observed at a concentration of 7 wt.%. of the polymer.

Molding mixture solutions were prepared at a temperature of 60 °C using an automatic high-speed stirrer in an ultrasonic bath. The ultrathin-fibrous nonwovens were obtained by the electrospinning technique on a single-capillary laboratory setup (developed in the N.N. Semenov Federal Research Center for Chemical Physics) with a capillary diameter of 0.1 mm, a voltage of 12 kV, a distance between the electrodes of 18 cm, and an electrical conductivity of 10 μS^−1^.

### 3.2. Methods

#### 3.2.1. Optical Microscopy

Optical microscopy was performed using a light microscope (Olympus BX3M-PSLED, Tokyo, Japan) in reflected light at magnifications of 50×, 100×, and 200×. The initial samples of non-woven matrices based on PLA/PHB blends and samples subjected to biological degradation using the Sturm test were analyzed.

#### 3.2.2. Fourier Transform Infrared Spectroscopy

The change in the chemical composition after biological degradation was studied using Fourier transform infrared spectroscopy (FTIR) on a IR-Fourier microscope (Lumos Bruker, Karlsruhe, Germany) using a macromodule by the method of frustrated total internal reflection (ATR) (ATR platinum diamond). The spectra were taken at a temperature of (23 ± 2) °C in the range of wave numbers of 4600–650 cm^−1^. The spectra were processed and the optical density of individual absorption bands was calculated using the Bruker OPUS software, version 7.0 (Karlsruhe, Germany).

#### 3.2.3. Thermogravimetric Analysis

Thermogravimetric analysis (TGA) was carried out using a synchronous thermal analysis device (TGA/DSC3+, Mettler Toledo, Greifensee, Switzerland) in the temperature range of +25… +800 °C at a heating rate of 10 deg/min in air (100.0 mL/min). For measurements, a 150 μL aluminum oxide crucible was used; the sample weight was 4–6 mg. The results were processed using the Star SW Lab Mettler software version 16.10 (Greifensee, Switzerland).

#### 3.2.4. Differential Scanning Calorimetry

The thermophysical properties of the materials were analyzed using a differential scanning calorimeter (DSC) (DSC 214 Polyma, NETZSCH-Gerätebau, Selb, Germany). The temperature scale and enthalpy of fusion are calibrated against standard samples of indium, zinc, and tin. The analysis was carried out in aluminum crucibles, Concavus NETZSCH-Gerätebau GmbH (Ø 5 mm, 30 μL), the sample weight was 4 ± 1 mg, and the heating/cooling rate was 10 °C/min. The following shooting mode was used: heating from +20 °C to +210 °C, cooling from +210 °C to +20 °C, and second heating from +20 °C to +210 °C. The thermograms (DSC curves) of the samples were normalized to a weight of (1 ± 0.1) g. The obtained data were processed using the Proteus NETZSCH software version 6.1 (Selb, Germany).

#### 3.2.5. Biological Degradation Analysis (Sturm Test)

The study of biological degradation was carried out according to GOST 32,427 (method B): test for biodegradation under the action of soil microorganisms on the release of carbon dioxide (Sturm test). A detailed description of the methodology is given in the paper [28]. To obtain the inoculum, soil was prepared according to Russian Standard GOST 9.060. In the process of biodegradation of materials by the microbiota of the inoculum, carbon dioxide was released. A 0.05 M Ba(OH)_2_ solution was used as an absorbent. The amount of released carbon dioxide was estimated by titration with a 0.1 M HCl solution by means of a Titrion automatic titration kit (OOO Econiks-Expert, Moscow, Russia). With a known initial content of total organic carbon in the samples, the biodegradation index was calculated. Analysis of the accumulation of carbon dioxide was carried out with an interval of 2–10 days. The biodegradation study was carried out for 60 days and the number of flasks for the study was 21: flasks 1–18 contained samples of the studied materials and 500 mL of inoculum (seed culture) and 19–21 contained only inoculum (zero control). Three replicates were performed for the biodegradation test.

#### 3.2.6. Oxidation by Ozon Medium (Ozonolysis)

The method for studying the reaction of ozone with non-woven matrices was carried out using an ozone generator connected to a gas chamber, in which the sample was placed; in this case, a gas flow with a certain ozone concentration and flow rate passed through the fixed sample (Figure 1). The reaction kinetics of ozone absorption by the samples was studied by changing the ozone concentration at the reactor outlet. In the device containing a detector (a spectrophotometric cuvette with a set long light wavelength), a change in the optical density of the absorption band at 254 nm was recorded, which indicated a change in the ozone concentration in the gas flow.

The amount of absorbed ozone by the sample was determined by the calculation method based on the volume of absorbed gas and the ability of the samples to react with free ozone, which entered the gas chamber where the sample was installed.

## 4. Conclusions

Creating mixtures of thermodynamically incompatible biopolymers is a key to creating effective biodegradable materials with a programmable self-destruction period in environmental or living systems, which is the most important direction to protect nature as a human habitat. In this work, this principle is demonstrated using the example of non-woven fibrous materials based on the blends of biopolymers: poly(3-hydroxybutyrate) and poly(lactic acid) obtained by electroforming. Changes in the structure of fibrous materials based on the blends of biodegradable polyesters during oxidative and biological degradation in a liquid inoculated medium were studied using structural–dynamic methods. Fibers of PLA and blends with PLA content from 30 to 90 wt.% had a regular cylindrical shape. At the same time, the structure of the material based on pure PHB and blends with 10 wt.% PLA were characterized by the presence of numerous defects in the form of ellipse-like thickenings. The detected effect indicates an influence of PLA on the technological properties of PHB during electroforming.

It was found that fibrous materials with a high PHB content (70–100 wt.%) underwent biodegradation to a significant extent. However, the addition of even 10 wt.% PLA significantly reduced the biodegradation rate of the PHB phase. When PHB content was below 50 wt.%, the biodegradation rate of the samples did not exceed 3% for 90 days. Apparently, products of PLA hydrolysis (lactic acid) poorly assimilated by soil microbiota were inhibitors of probiotic development.

The supramolecular structure of polymers in the blends underwent significant changes after prolonged exposure in the inoculum. DSC data indicates a decrease in the melting temperature of the polymers in the blends after biodegradation. The crystallinity (enthalpy) decreased significantly at both the first and the second heating.

The FTIR-spectroscopy showed that changes in the molecular structure of polymers were also significant: the absorption bands characteristic for PLA remained practically unchanged, while the absorption bands characteristic for PHB bands almost completely disappeared after the biodegradation test.

Our numerous studies did not reveal any significant intermolecular interaction in the blends of PHB and PLA. The behavior of blended polymers is determined by the individual characteristics of each component and depends on the concentration of the less stable polymer. For example, the dependence of the thermal degradation of blends on the content of less heat-resistant PHB was observed.

In the study of oxidative degradation of fibrous materials by ozone, it was shown that blended materials had an increased rate of oxidation by ozone compared to homopolymers. This appears to be due to the lower packing density of macro-chains in the amorphous regions of the blended samples. A two-step kinetic mechanism of ozonation including the formation of side groups and subsequent degradation of the polymer chain was demonstrated. It was found that the activation energy of ozonation depends on the ratio of polymers in the blend: the maximum values were in the phase-inversion region of 30–70 wt.% of PLA. It was shown that the rate of ozonation is higher for PLA and the blends with a high PLA content.

It was found that the PLA/PHB blends demonstrated relatively high rates of biodegradation and oxidation, which is their advantage when creating completely biodegradable materials and products. High porosity, moderate hydrophobicity, and good environmental biodegradability make these materials competitive as environmentally friendly sorbents, as well as agricultural and packaging materials. However, the biomedical application of PLA/PHB materials will be problematic, since the hydrolysis of PLA can release lactic acid, which causes inflammatory reactions in living tissues.

The fibrous materials based on the blend of 30PLA/70PHB wt.% have the most optimal structural–dynamic parameters and the ability for biological and oxidative degradation.

## Figures and Tables

**Figure 1 ijms-24-07979-f001:**
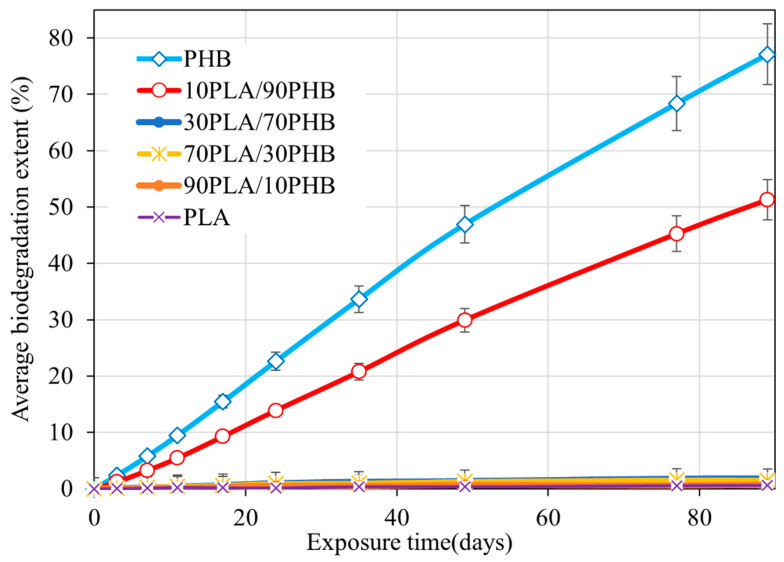
Dependence of the degree of biodegradation of PLA/PHB fibrous matrices in the inoculum on the exposure time (Sturm test).

**Figure 2 ijms-24-07979-f002:**
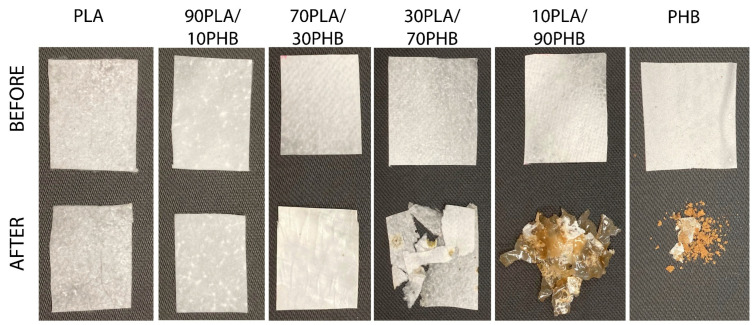
Appearance of PLA/PHB non-woven fibrous matrices before and after 90 days of biodegradation (Sturm test).

**Figure 3 ijms-24-07979-f003:**
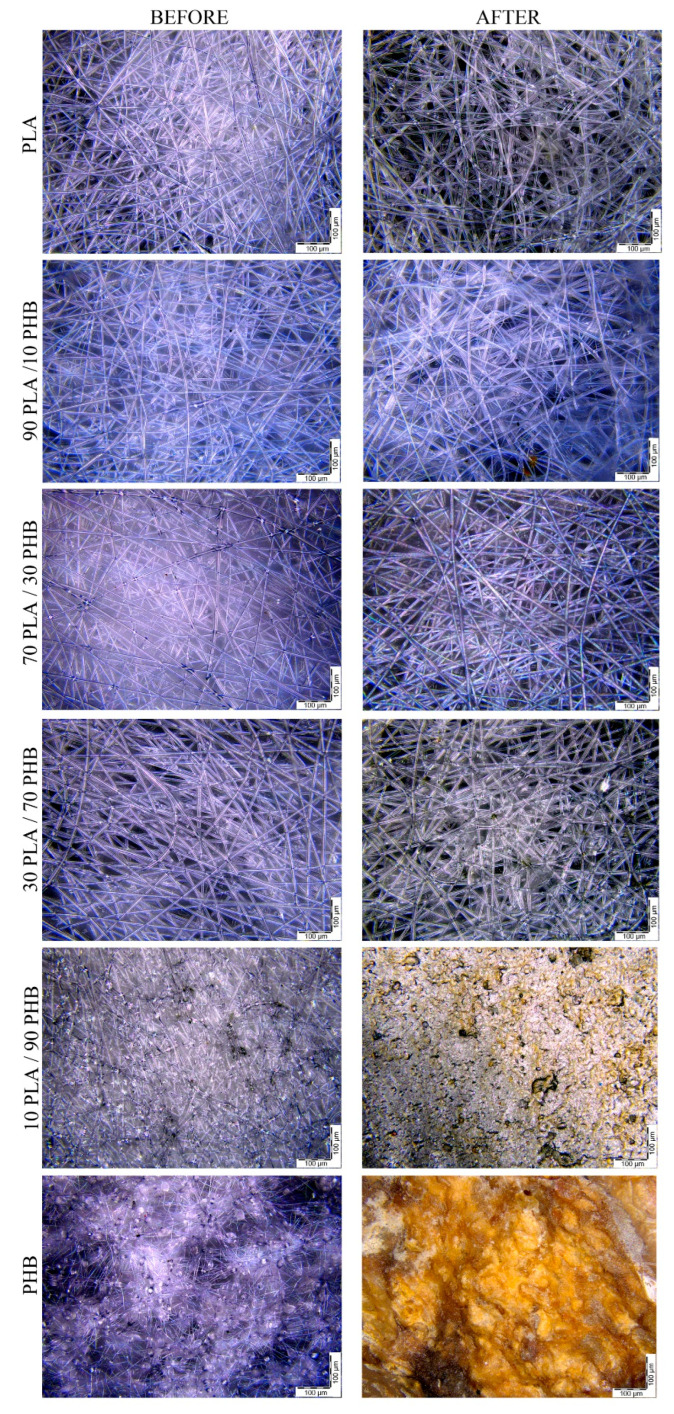
Microphotographs of PLA/PHB non-woven fibrous matrices before and after 90 days biodegradation (Sturm test).

**Figure 4 ijms-24-07979-f004:**
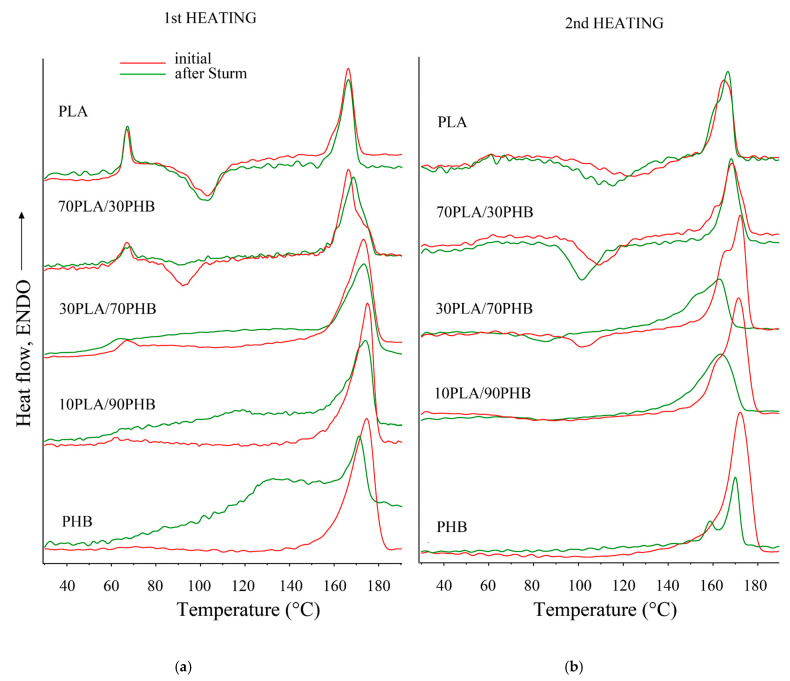
DSC curves of the first (**a**) and second (**b**) heating of PLA/PHB fibrous matrices for the initial samples (red curves) and the samples after the Sturm test (green curves).

**Figure 5 ijms-24-07979-f005:**
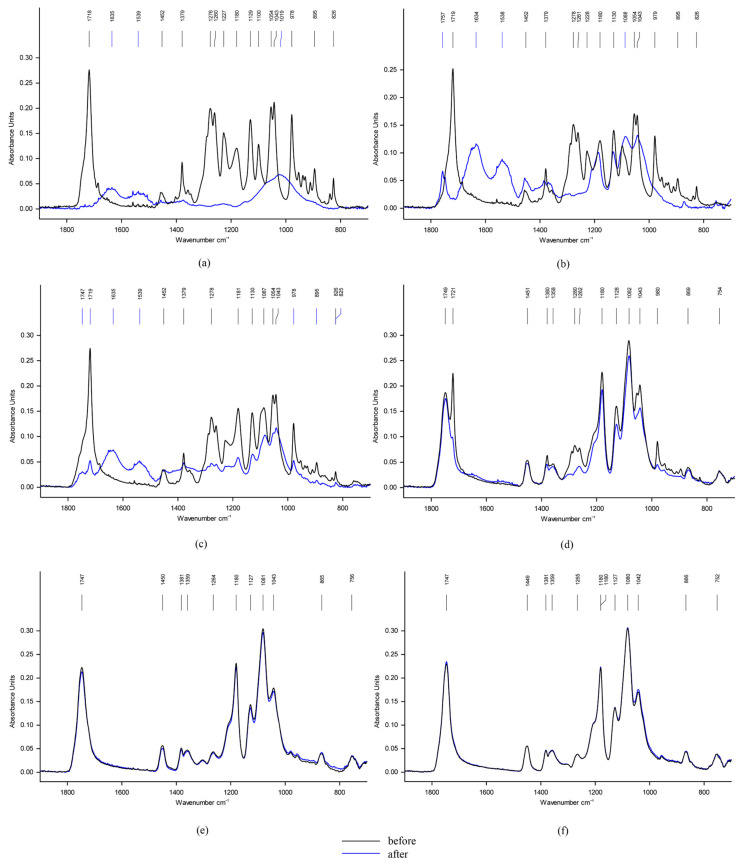
FTIR spectra of the fibrous matrices based on PLA/PHB blends with 0 (**a**), 10 (**b**), 30 (**c**), 70 (**d**), 90 (**e**), and 100 (**f**) wt.% of PLA for the initial samples (black curves) and the samples after the Sturm test (blue curves).

**Figure 6 ijms-24-07979-f006:**
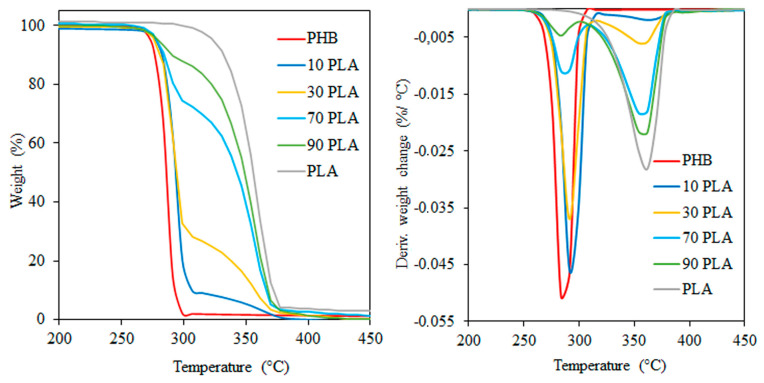
Thermograms of thermogravimetric analysis of PLA/PHB nonwoven fibrous materials depending on the composition.

**Figure 7 ijms-24-07979-f007:**
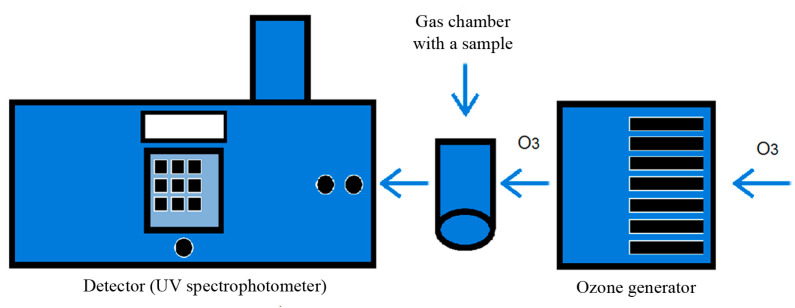
Scheme of the experimental setup for ozonation of samples.

**Figure 8 ijms-24-07979-f008:**
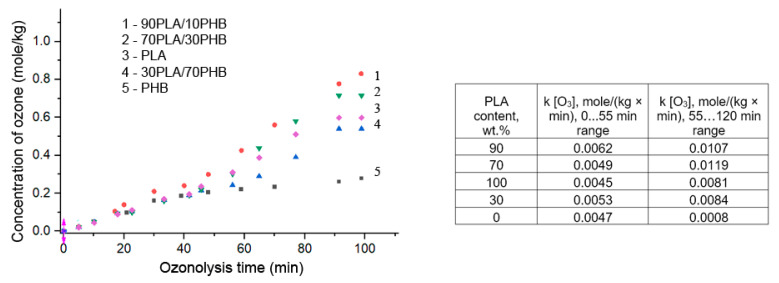
Ozone absorption by PLA/PHB ultrafine fibers (wt.%) during the exposure time.

**Table 1 ijms-24-07979-t001:** Melting temperature (T_m_) and enthalpy of fusion (ΔH_m_) of the polymers for 1st and 2nd heating before and after Sturm test.

PLA Content (wt.%)	1st Heating				2nd Heating			
	before		after		before		after	
	T_m_ (°C)	ΔH_m_ (J/g)	T_m_ (°C)	ΔH_m_ (J/g)	T_m_ (°C)	ΔH_m_ (J/g)	T_m_ (°C)	ΔH_m_ (J/g)
0	175.3 ± 0.2	83 ± 2	168.3 ± 0.5	70 ± 5	172.6 ± 0.3	78 ± 2	170.0 ± 0.6	25 ± 8
10	175.0 ± 0.3	76 ± 2	174.1 ± 0.4	50 ± 4	171.5 ± 0.1	73 ± 1	164.3 ± 0.4	52 ± 6
30	173.9 ± 0.1	63 ± 2	174.1 ± 0.2	52 ± 3	172.8 ± 0.1	62 ± 2	163.0 ± 0.4	41 ± 3
70	166.5 ± 0.3	44 ± 2	168.6 ± 0.2	41 ± 2	168.7 ± 0.3	35 ± 2	168.0 ± 0.3	34 ± 1
90	167.3 ± 0.4	35 ± 3	166.3 ± 0.3	34 ± 1	168.7 ± 0.1	32 ± 3	168.5 ± 0.1	36 ± 3
100	166.5 ± 0.2	31 ± 1	166.4 ± 0.1	32 ± 2	164.8 ± 0.2	33 ± 2	166.7 ± 0.2	35 ± 1

## Data Availability

Data is unavailable due to privacy or ethical restrictions.

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
