# Peer review of "Biological and Oxidative Degradation of Ultrathin-Fibrous Nonwovens Based on Poly(lactic Acid)/Poly(3-Hydroxybutyrate) Blends"

_ijms, 2023, doi:10.3390/ijms24097979_

Round 1

Reviewer 1 Report

This paper presents different polymer matrices based on PLA and PHB obtained in different ratios by electrospinning and their biological degradation.

Some minor observations:

1. The authors claim that they developed ultrathin fibrous polymeric matrices. Have thickness measurements been made and compared with other examples in the literature?

2. Figure 4: the scale of the imagines is unclear. A higher text font for visibility is welcome.

3. Table 1 and the one from figure 8: change comma with dot for the values

The Conclusion section is described in a very detailed manner. This section could be split into Discussion and further Conclusion or be included in Section 3.

Author Response

Dear reviewer!

Thank you very much for your valuable comments. They improved the quality of the article. 

Sincerely, Dr. A. Olkhov.

Reviewer 2 Report

Olkhov et al. fabricated a series of ultrathin-fibrous nonwovens with different blending ratios of poly(lactic acid) and poly(3-hydroxybutyrate), and mainly explored their biological and oxidative degradation. Some major revisions should be conducted before publication.

1. The Abstract section should be presented in a better and clear way. For instance, some important result data like degradation performance are suggested to be presented in a much more detailed manner.

2. Please state the reasons why both PHA and PLA were chosen in this study. What are the merits and advantages of PHA and PLA compared with some other biopolymers like PCL, PLGA, etc.?

3. The merits of electrospinning technique should be outlined in the Introduction section, and some recent works about the innovative electrospinning like European Polymer Journal, 2023, 186, 111863, https://doi.org/10.1016/j.eurpolymj.2023.111863 and ACS Applied Materials & Interfaces, 2022, 14(14), 15911-15926, https://doi.org/10.1021/acsami.1c24131 are suggested to be discussed.

4. Please state the reasons why chloroform was utilized as solvent, and 7% polymeric concentration was chosen for electrospinning. Moreover, how did the authors choose the parameters of electrospinning? Do they conduct any preliminary experiments?

5. Figure 2: How many replicates were performed for the degree of biodegradation test? The error bars should be added.

6. The authors didn’t mention the fiber diameters of as-prepared PHA/PLA ultrathin-fibrous nonwovens. Did the PHA/PLA weight ratio affect the fiber diameter? The related descriptions are suggested to be added.

7. The scale bars are not clear in the Figure 4, which should be redrawn.

8. How about the mechanical stability of as-prepared PHA/PLA ultrathin-fibrous nonwovens? Did the PHA/PLA weight ratio affect the mechanical stability?

9. Some more discussion should be added. The potential application field is suggested to be added. Moreover, the limitation and challenges should also be discussed.

10. The grammar and writing should be improved in the whole manuscript.

Author Response

Dear reviewer! Thank you very much for your valuable comments. They improved the quality of the article. My answers are in the attached document.

Sincerely, Dr. A. Olkhov.

Round 2

Reviewer 2 Report

The authors have addressed the reviewer's comments well.